# AccCtr: Accelerating Training-Free Control For Text-to-Image Diffusion Models

## Abstract

In training-free Conditional Diffusion Models (CDMs), the sampling process is steered by the gradient of the loss $\mathcal{E}(\mathbf{y}, \mathbf{z}, C_\psi)$, which assesses the gap between the guidance $\mathbf{y}$ and the condition extracted from the intermediate outputs. Here the condition extraction network $C_\psi(\cdot)$, which could be a segmentation or depth estimation network, is pre-trained for training-free purpose. However, existing methods often require small guidance steps, leading to longer sampling times. We introduce an alternative maximization framework to scrutinize training-free CDMs that tackles slow sampling. Our framework pinpoints manifold deviation as the key factor behind the sluggish sampling. More iterations are needed for the sampling process to closely follow the image manifold and reach the target conditions, as the loss gradient doesn't provide sufficient guidance for larger steps. To improve this, we suggest retraining the condition extraction network $C_\psi(\cdot)$ to refine the loss's guidance, thereby introducing our AccCtr. This retraining process is simple, and integrating AccCtr into current CDMs is a seamless task that does not impose a significant computational burden. Extensive testing has demonstrated that AccCtr significantly boosts performance, offering superior sample quality and faster generation times across a variety of conditional generation tasks.

## 1 Introduction

Over the past few years, diffusion models (Sohl-Dickstein et al., 2015; Song & Ermon, 2019; Ho et al., 2020; Song et al., 2021b) have achieved significant success in generative tasks like image generation Nichol & Dhariwal (2021); Song & Ermon (2020); Song et al. (2021a), image inpainting Chung et al. (2023), super-resolution Saharia et al. (2023), image editing Choi et al. (2021), thanks to their strong expressive and re-editing capabilities.

Conditional diffusion models generally employ two techniques: classifier-guided Dhariwal & Nichol (2021) and classifier-free Ho & Salimans (2021a) diffusion models. Despite their effectiveness, these methods encounter challenges related to learning cost and model generality, as they require additional training and data for conditional generation. Recent advances Chung et al. (2022); Zhu et al. (2023); Yu et al. (2023); Bansal et al. (2024); Yang et al. (2024b) have addressed these issues by developing training-free methods that leverage the differential loss guidance during the denoising process.

These training-free methods, though they avoid extra training, demand fine-tuned guidance steps for accuracy, which extends sampling times. This is mainly because the tangent space defined by the differential loss can only approximate a local image manifold area. If the starting point is remote from the target, multiple manifolds are needed to span the gap. Thus, more iterations are crucial for the denoising process to navigate the manifold's curvature and reach the target condition effectively. Current approaches Chung et al. (2023); Yu et al. (2023); Bansal et al. (2023) often use small loss-guided steps to ensure precision, which can considerably slow down the process. However, Yang et al. (2024b) has made significant progress by enabling larger guidance steps through optimization, thus improving algorithm efficiency.

Unlike Yang et al. (2024b) using optimization to constrain the guidance steps to remain within the boundaries of the intermediate data, we improve the efficiency with a alternative maximization framework that simplifies the sampling in training-free CDMs to optimizing two objectives: $\log p(\mathbf{z}_0)$ for unconditional generation and $\log p(\mathbf{y}|\mathbf{z}_0)$ for the conditional generation. Here, $\mathbf{z}_0$ represents the denoised image of the diffusion model at time step 0. We denote the image manifold consisting of $\mathbf{z}_0$ as $M_0$. This new interpretation guides us to streamline sampling by reducing the optimization steps necessary for each objective. Our further study reveals that reducing the optimization steps for $\log p(\mathbf{z}_0)$ is straightforward, but not so for $\log p(\mathbf{y}|\mathbf{z}_0)$. Taking the value of a well-trained model $\mathbf{s}(\mathbf{z}_t)$, we can estimate the denoised image $\mathbf{z}_{0|t}$, *i.e.* the projection of $\mathbf{z}_t$ on the manifold $M_0$, in one step. However, maximizing $\log p_{\mathbf{y}}(\mathbf{z}_{0|t})$ involves the gradient of $\mathcal{E}(\mathbf{y}, \mathbf{z}_{0|t}, \boldsymbol{C_\psi})$ and requires multiple steps for gradient descent to reach the final outcome.

To reduce the maximization steps needed for $\log p(\mathbf{y}|\mathbf{z}_{0|t})$, we propose retaining the condition extraction network $\boldsymbol{C_\psi}(\cdot)$ to enhance its ability so that the gradient of $\mathcal{E}(\mathbf{y}, \mathbf{z}_{0|t}, \boldsymbol{C_\psi})$ provides a more accurate direction for larger steps. Consequently, it is logical to retrain the network $\boldsymbol{C_\psi}(\cdot)$ with two distinct objectives. The first is to ensure that $\boldsymbol{C_\psi}(\mathbf{z}_{0|t})$ effectively extracts the necessary conditions from $\mathbf{z}_{0|t}$. The second is to adjust the gradient of $\mathcal{E}(\mathbf{y}, \mathbf{z}_{0|t}, \boldsymbol{C_\psi})$ so that it provides accurate guidance for larger steps.

In summary, our contributions are fourfold: 1. We introduce a novel maximization framework that provides insights into the analysis of training-free CDMs. 2. We identify the key bottleneck in the generation speed of current training-free CDMs using this framework. 3. We propose a loss to retrain the condition extraction network to address this bottleneck. 4. Our model outperforms previous models in efficiency and sample quality.

## 2 Related work

Conditional Diffusion Models (CDMs) are typically divided into two categories: training-required and training-free. A key aspect of both types of models is the estimation of the conditional score $\nabla_{\mathbf{z}_t} \log p(\mathbf{z}_t, \mathbf{y})$ or its component $\nabla_{\mathbf{z}_t} \log p(\mathbf{y}|\mathbf{z}_t)$, which is derived from the relationship $\nabla_{\mathbf{z}_t} \log p(\mathbf{z}_t, \mathbf{y}) = \nabla_{\mathbf{z}_t} \log p(\mathbf{z}_t) + \nabla_{\mathbf{z}_t} \log p(\mathbf{y}|\mathbf{z}_t)$.

**Training-required CDMs** are categorized into two branches. The first one is the classifier-guided diffusion mode (Dhariwal & Nichol, 2021), training a time-dependent classifier denoted as $p_\phi(\mathbf{y}|\mathbf{z}_t, t)$ to approximate the posterior probability $p(\mathbf{y}|\mathbf{z}_t)$. Consequently, we have $\nabla_{\mathbf{z}_t} \log p(\mathbf{z}_t, \mathbf{y}) = \nabla_{\mathbf{z}_t} \log p(\mathbf{z}_t) + \nabla_{\mathbf{z}_t} \log p_\phi(\mathbf{y}|\mathbf{z}_t, t)$, where the first term represents the unconditional score function, while the second term signifies the adjustment that converts the unconditional score into a conditional one. The other one is the classifier-free diffusion model (Ho & Salimans, 2021b). This approach employs a neural network to approximate the conditional score $\nabla_{\mathbf{z}_t} \log p(\mathbf{z}_t, \mathbf{y})$. Notable examples include Stable Diffusion (Rombach et al., 2022b), ControlNet (Zhang et al., 2023), and ControlNet++ (Ming Li, 2024), ControlNeXt (Peng et al., 2024), and AnyControl (Sun et al., 2024). These models are great at creating realistic images but require more data and training time.

**Training-free CDMs** eliminates classifier training by defining a loss $\mathcal{E}(\mathbf{y}, \mathbf{z}_{0|t}, \boldsymbol{C_\psi})$ and using its gradient to approximate the conditional score $\nabla_{\mathbf{z}_t} \log p(\mathbf{y}|\mathbf{z}_t)$. In the literature, researchers devised various strategies to improve the conditional score estimation. MCG (Chung et al., 2022) addresses solver deviations with a correction term. DPS (Chung et al., 2023) integrates diffusion sampling with manifold constraints for better noise handling. FreeDoM (Yu et al., 2023) uses a Time-Travel Strategy for robust generation. UGD (Bansal et al., 2024) and DiffPIR (Zhu et al., 2023) guide clean samples $\mathbf{z}_0$ to intermediate manifolds $\mathbf{z}_t$. LGD (Song et al., 2023) uses Monte Carlo sampling for estimation refinement. MPGD (He et al., 2024) and DSG (Yang et al., 2024b) apply guidance within data manifolds, with DSG providing a closed-form solution. These approaches often require around 100 sampling steps for quality generation, contrasting with the typically less than 20 steps needed by training-required CDMs.

We in this paper delve into the rationale behind the increased sampling steps required for training-free CDMs and propose a strategy to enhance their efficiency.

## 3 PRELIMINARIES

Diffusion models (Yang et al., 2024a) are understood through various lenses, such as the Denoising Diffusion Probabilistic Model (DDPM) (Ho et al., 2020), Score-Matching Langevin Dynamics (SMLD) (Song & Ermon, 2019), and Stochastic Differential Equations (SDE) (Song et al., 2021b). This section offers essential background into DDPM related to our method.

### 3.1 DIFFUSION AND MAXIMIZATION

Diffusion models are represented as: $p_{\boldsymbol{\theta}}(\mathbf{z}_0) = \int p_{\boldsymbol{\theta}}(\mathbf{z}_{0:T}) \, d\mathbf{z}_{1:T}$, where $\mathbf{z}_1, \ldots, \mathbf{z}_T$ are latent variables of the same dimension as the data $\mathbf{z}_0 \sim q(\mathbf{z}_0)$. The joint distribution $p_{\boldsymbol{\theta}}(\mathbf{z}_{0:T})$ is defined by a Markov chain with Gaussian transitions starting from $\mathbf{z}_T \sim \mathcal{N}(\mathbf{z}_T; \mathbf{0}, \boldsymbol{I})$:

$$p_{\boldsymbol{\theta}}(\mathbf{z}_{0:T}) \coloneqq p(\mathbf{z}_T) \prod_{t=1}^{T} p_{\boldsymbol{\theta}}(\mathbf{z}_{t-1}|\mathbf{z}_t), \qquad p_{\boldsymbol{\theta}}(\mathbf{z}_{t-1}|\mathbf{z}_t) \coloneqq \mathcal{N}(\mathbf{z}_{t-1}; \boldsymbol{\mu}_{\boldsymbol{\theta}}(\mathbf{z}_t, t), \boldsymbol{\Sigma}_{\boldsymbol{\theta}}(\mathbf{z}_t, t)) \quad (1)$$

The forward diffusion process, gradually introducing Gaussian noise to the data, is defined by a Markov chain with a predetermined variance schedule $\beta_1, \ldots, \beta_T$:

$$q(\mathbf{z}_{1:T}|\mathbf{z}_0) \coloneqq \prod_{t=1}^{T} q(\mathbf{z}_t|\mathbf{z}_{t-1}), \qquad q(\mathbf{z}_t|\mathbf{z}_{t-1}) \coloneqq \mathcal{N}(\mathbf{z}_t; \sqrt{1-\beta_t}\mathbf{z}_{t-1}, \beta_t \boldsymbol{I}) \quad (2)$$

Let $M_0$ represent the image manifold generated by the diffusion model. This process allows for sampling $\mathbf{z}_t$ at any time step $t$ and deriving its projection onto $M_0$ in closed form:

$$q(\mathbf{z}_t|\mathbf{z}_0) = \mathcal{N}(\mathbf{z}_t; \sqrt{\bar{\alpha}_t}\mathbf{z}_0, (1-\bar{\alpha}_t)\boldsymbol{I}), \quad \text{where} \quad \bar{\alpha}_t \coloneqq \prod_{t=1}^{T} \alpha_s, \alpha_t \coloneqq 1 - \beta_t \quad (3)$$

$$\Leftrightarrow \qquad \mathbf{z}_t = \sqrt{\bar{\alpha}_t}\mathbf{z}_0 + \sqrt{(1-\bar{\alpha}_t)}\boldsymbol{\epsilon}, \quad \text{where} \quad \boldsymbol{\epsilon} \sim \mathcal{N}(\mathbf{0}, \boldsymbol{I}) \quad (4)$$

$$\Leftrightarrow \qquad \mathbf{z}_0 = \frac{1}{\sqrt{\bar{\alpha}_t}}\mathbf{z}_t - \frac{\sqrt{(1-\bar{\alpha}_t)}}{\sqrt{\bar{\alpha}_t}}\boldsymbol{\epsilon}(\mathbf{z}_t) \qquad \Leftrightarrow \qquad \mathbf{z}_0 = \frac{1}{\sqrt{\bar{\alpha}_t}}\mathbf{z}_t + \frac{(1-\bar{\alpha}_t)}{\sqrt{\bar{\alpha}_t}}\boldsymbol{s}(\mathbf{z}_t) \quad (5)$$

Here $\boldsymbol{\epsilon}(\mathbf{z}_t)$ denote the noised contained in $\mathbf{z}_t$ and the score function $\boldsymbol{s}(\mathbf{z}_t) \coloneqq \nabla_{\mathbf{z}_t} \log p(\mathbf{z}_t)$ satisfying $\boldsymbol{\epsilon}(\mathbf{z}_t) = -\sqrt{1-\bar{\alpha}_t}\boldsymbol{s}(\mathbf{z}_t)$ due to Tweedie's formula (Efron, 2011). Let $\tilde{\boldsymbol{\mu}}(\mathbf{z}_t, \mathbf{z}_0, t) \coloneqq \frac{\sqrt{\bar{\alpha}_{t-1}}\beta_t}{1-\bar{\alpha}_t}\mathbf{z}_0 + \frac{\sqrt{\alpha_t}(1-\bar{\alpha}_{t-1})}{1-\bar{\alpha}_t}\mathbf{z}_t$ and $\tilde{\beta}_t \coloneqq \frac{1-\bar{\alpha}_{t-1}}{1-\bar{\alpha}_t}\beta_t$, $q(\mathbf{z}_{t-1}|\mathbf{z}_t, \mathbf{z}_0)$ can be written as

$$q(\mathbf{z}_{t-1}|\mathbf{z}_t, \mathbf{z}_0) = \mathcal{N}(\mathbf{z}_{t-1}; \tilde{\boldsymbol{\mu}}(\mathbf{z}_t, \mathbf{z}_0, t), \bar{\beta}_t \boldsymbol{I}), \quad (6)$$

$$\Leftrightarrow \qquad \mathbf{z}_{t-1} = \frac{\sqrt{\bar{\alpha}_{t-1}}\beta_t}{1-\bar{\alpha}_t}\mathbf{z}_0 + \frac{\sqrt{\alpha_t}(1-\bar{\alpha}_{t-1})}{1-\bar{\alpha}_t}\mathbf{z}_t + \sqrt{\bar{\beta}_t}\boldsymbol{\epsilon} \quad (7)$$

By defining $\boldsymbol{s}_{\boldsymbol{\theta}}(\mathbf{z}_t)$ as the neural network designed to approximate the score function $\boldsymbol{s}(\mathbf{z}_t)$ and substituting it into Equation (5), we obtain $\hat{\mathbf{z}}_{0|t-1}$, an estimation for $\mathbf{z}_0$ according to $\mathbf{z}_{t-1}$.

$$\hat{\mathbf{z}}_{t-1} = \frac{\sqrt{\bar{\alpha}_{t-1}}\beta_t}{1-\bar{\alpha}_t}\hat{\mathbf{z}}_0^{(t)} + \frac{\sqrt{\alpha_t}(1-\bar{\alpha}_{t-1})}{1-\bar{\alpha}_t}\hat{\mathbf{z}}_t + \sqrt{\bar{\beta}_t}\boldsymbol{\epsilon} \quad (8)$$

$$\hat{\mathbf{z}}_{0|t-1} = \frac{1}{\sqrt{\bar{\alpha}_t}}\hat{\mathbf{z}}_t + \frac{(1-\bar{\alpha}_t)}{\sqrt{\bar{\alpha}_t}}\boldsymbol{s}_{\boldsymbol{\theta}}(\hat{\mathbf{z}}_t) \quad (9)$$

We thus confirm that $\hat{\mathbf{z}}_{0|t}$ is the projection of $\hat{\mathbf{z}}_t$ on the image manifold $M_0$, and the sequence $\{\hat{\mathbf{z}}_{0|t}\}$ maximizes $\log p(\hat{\mathbf{z}}_{0|t})$. Hence, **we view Equations (8)(9) as the solver for maximizing** $\log p(\mathbf{z}_0)$ **on the manifold** $M_0$, **which includes all** $\mathbf{z}_0$ **generated by the diffusion model.**

### 3.2 CONDITIONAL DIFFUSION

Conditional diffusion models employ the conditional score $\boldsymbol{s}(\mathbf{z}_t, \mathbf{y}) \coloneqq \nabla_{\mathbf{z}_t} \log p(\mathbf{z}_t, \mathbf{y})$ as a substitute for $\boldsymbol{s}(\mathbf{z}_t)$ in Equation (9), enabling the generation of images conditioned on $\mathbf{y}$. This function is articulated via Bayes' theorem as follows: $\boldsymbol{s}(\mathbf{z}_t, \mathbf{y}) = \boldsymbol{s}(\mathbf{z}_t) + \nabla_{\mathbf{z}_t} \log p(\mathbf{y}|\mathbf{z}_t)$. To sidestep training, a practical approach is to use an energy function, defined as: $\log p(\mathbf{y}|\mathbf{z}_t) =$

---

**Algorithm 1** Alternative Maximization Sampling

---

**Require:** The iteration number $J$, the unconditional diffusion count $N$ for solving $p(\mathbf{z}_{0|t})$ and the conditional correction count $M$ for solving $p_{\boldsymbol{y}}(\mathbf{z}_{0|t})$. The time reversal step $K$.

**Ensure:** $\hat{\mathbf{z}}_{JN} \sim \mathcal{N}(\mathbf{0}, \boldsymbol{I})$, and $\hat{\mathbf{z}}_{0|JN} \leftarrow \sqrt{\bar{\alpha}_{JN}}^{-1}(\hat{\mathbf{z}}_{JN} + (1 - \bar{\alpha}_{JN})s_{\boldsymbol{\theta}}(\hat{\mathbf{z}}_{JN}))$

1: **for** $j = J, \ldots, 1$ **do**

2:     **for** $n = 0, \ldots, N - 1$ **do**

3:         $t \leftarrow jN - n$

4:         $\hat{\mathbf{z}}_{t-1} \leftarrow \frac{\sqrt{\bar{\alpha}_{t-1}}\beta_t}{1 - \bar{\alpha}_t}\hat{\mathbf{z}}_{0|t} + \frac{\sqrt{\alpha_t}(1 - \bar{\alpha}_{t-1})}{1 - \bar{\alpha}_t}\hat{\mathbf{z}}_t + \sqrt{\bar{\beta}_t}\boldsymbol{\epsilon}$

5:         $\hat{\mathbf{z}}_{0|t-1} \leftarrow \frac{1}{\sqrt{\bar{\alpha}_{t-1}}}\hat{\mathbf{z}}_{t-1} + \frac{(1 - \bar{\alpha}_{t-1})}{\sqrt{\bar{\alpha}_{t-1}}}s_{\boldsymbol{\theta}}(\hat{\mathbf{z}}_{t-1})$

6:     **end for**

7:     $t \leftarrow (j - 1)N$

8:     **for** $m = 0, \ldots, M - 1$ **do**

9:         $\hat{\mathbf{z}}_{K|t}^{(m)} \leftarrow \sqrt{\bar{\alpha}_K}\hat{\mathbf{z}}_{0|t}^{(m)} + \sqrt{(1 - \bar{\alpha}_K)}\boldsymbol{\epsilon}$         ▷ Adding noisy to $\hat{\mathbf{z}}_{0|t}^{(m)}$.

10:        $\hat{\mathbf{z}}_{0|t}^{(m)} \leftarrow \frac{1}{\sqrt{\bar{\alpha}_t}}\hat{\mathbf{z}}_{K|t}^{(m)} + \frac{(1 - \bar{\alpha}_K)}{\sqrt{\bar{\alpha}_K}}s_{\boldsymbol{\theta}}(\hat{\mathbf{z}}_{K|t}^{(m)})$     ▷ Estimating a new $\hat{\mathbf{z}}_{0|t}^{(m)}$.

11:        $\hat{\mathbf{z}}_{0|t}^{(m+1)} \leftarrow \hat{\mathbf{z}}_{0|t}^{(m)} - \lambda\nabla_{\hat{\mathbf{z}}_{0|t}^{(m)}}\mathcal{E}(\mathbf{y}, \hat{\mathbf{z}}_{0|t}^{(m)}, \boldsymbol{C}_{\boldsymbol{\psi}})$

12:     **end for**

13:     $\hat{\mathbf{z}}_{0|t} \leftarrow \hat{\mathbf{z}}_{0|t}^{(M)}$

14: **end for**

---

$-\lambda\mathcal{E}(\mathbf{y}, \mathbf{z}_{0|t}, \boldsymbol{C}_{\boldsymbol{\psi}})$, where $\mathbf{z}_{0|t} = \sqrt{\bar{\alpha}_t}^{-1}(\mathbf{z}_t + (1 - \bar{\alpha}_t))s_{\boldsymbol{\theta}}(\mathbf{z}_t)$. In this expression, $\lambda$ is a positive parameter. Consequently, Equations (8)(9) can be restructured accordingly.

$$\hat{\mathbf{z}}_{t-1} = \frac{\sqrt{\bar{\alpha}_{t-1}}\beta_t}{1 - \bar{\alpha}_t}\hat{\mathbf{z}}_{0|t} + \frac{\sqrt{\alpha_t}(1 - \bar{\alpha}_{t-1})}{1 - \bar{\alpha}_t}\hat{\mathbf{z}}_t + \sqrt{\bar{\beta}_t}\boldsymbol{\epsilon} \tag{10}$$

$$\hat{\mathbf{z}}_{0|t-1}' = \frac{1}{\sqrt{\bar{\alpha}_{t-1}}}\hat{\mathbf{z}}_{t-1} + \frac{(1 - \bar{\alpha}_{t-1})}{\sqrt{\bar{\alpha}_{t-1}}}s_{\boldsymbol{\theta}}(\hat{\mathbf{z}}_{t-1}) \tag{11}$$

$$\hat{\mathbf{z}}_{0|t-1} = \hat{\mathbf{z}}_{0|t-1}' - \lambda\frac{(1 - \bar{\alpha}_{t-1})}{\sqrt{\bar{\alpha}_{t-1}}}\nabla_{\hat{\mathbf{z}}_{t-1}}\mathcal{E}(\mathbf{y}, \hat{\mathbf{z}}_{t-1}, \boldsymbol{C}_{\boldsymbol{\psi}}) \tag{12}$$

Further, given Equation (4), we have $\nabla_{\hat{\mathbf{z}}_{t-1}}\mathcal{E}(\mathbf{y}, \hat{\mathbf{z}}_{t-1}, \boldsymbol{C}_{\boldsymbol{\psi}}) = \sqrt{\bar{\alpha}_{t-1}}\nabla_{\hat{\mathbf{z}}_{0|t-1}'}\mathcal{E}(\mathbf{y}, \hat{\mathbf{z}}_{0|t-1}', \boldsymbol{C}_{\boldsymbol{\psi}})$. Putting this into Equation (12), we conclude that it operates as a gradient descent step for $\mathcal{E}(\mathbf{y}, \hat{\mathbf{z}}_{0|t-1}', \boldsymbol{C}_{\boldsymbol{\psi}})$. In contrast, Equations (10)(11) serve as a solver to maximize $p(\hat{\mathbf{z}}_{0|t}')$. **Essentially, these equations alternately maximize the two objectives** $\log p(\hat{\mathbf{z}}_0')$ **and** $\log p(\mathbf{y}|\hat{\mathbf{z}}_{0|t})$ **on the image manifold** $M_0$ **with each step focusing on one objective.** Thus, the sequence $\{\hat{\mathbf{z}}_{0|t}'\}$ maximizes $\log p(\hat{\mathbf{z}}_{0|t}')$, while the sequence $\{\hat{\mathbf{z}}_{0|t}\}$ maximizes $\log p_{\mathbf{y}}(\hat{\mathbf{z}}_{0|t})$.

## 4 ALTERNATIVE MAXIMIZATION FOR CONDITIONAL DIFFUSION

In this section, we frame the conditional diffusion process as an alternating maximization of two objectives: $p(\mathbf{z}_0)$ and $p(\mathbf{y}|\mathbf{z}_0)$. This insight helps us understand why training-free CDMs require more sampling steps and leads to a strategy for speeding up the process.

### 4.1 THE LOCAL MAXIMA CHARACTERISTICS OF $p(\mathbf{z}_0)$ AND $p(\mathbf{y}|\mathbf{z}_0)$

The marginal distribution $p(\mathbf{z}_0)$ peaks at natural images, and the condition extraction function $\boldsymbol{C}_{\boldsymbol{\psi}}(\cdot)$ is tailored for such images. The conditional distribution $p(\mathbf{y}|\mathbf{z}_0)$ reaches its peak when $\mathbf{y}$ matches $\mathbf{z}_0$, with $p(\mathbf{y}|\mathbf{z}_0) \geq p(\mathbf{y}|\mathbf{z})$ for neighboring images $\mathbf{z} \neq \mathbf{z}_0$. Therefore, $p(\mathbf{y}|\mathbf{z}_0)$ attains its maximum where $p(\mathbf{z}_0)$ is locally maximized. Consequently, the local maxima of $p(\mathbf{y}|\mathbf{z}_0)$ form a subset of the local maxima of $p(\mathbf{z}_0)$. In other words, wherever $p(\mathbf{z}_0)$ is locally maximized, $p(\mathbf{y}|\mathbf{z}_0)$ is also likely to achieve a local maximum, provided $\mathbf{y}$ describes $\mathbf{z}_0$. This relationship emphasizes the role of the conditional distribution in guiding the generative process toward images that not only align with the natural image distribution but also closely match the specified conditions.

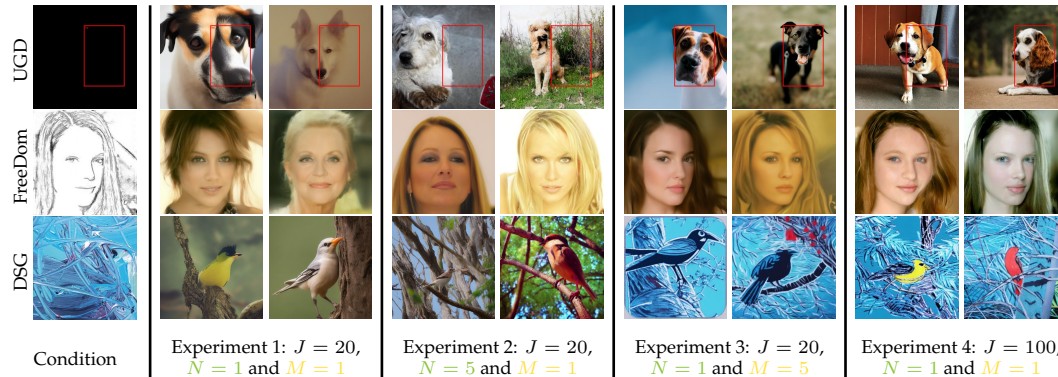

Figure 1: Analysis of the Impact of Iteration Counts: Total $J$, Unconditional $N$ and Conditional $M$. From top to bottom, each row shows the outcomes of FreeDoM (Yu et al., 2023), DSG (Yang et al., 2024b), and UGD (Bansal et al., 2024) under conditions of edge, style, and bounding box control. Four experiments were conducted in total. Observations reveal that the first two setups failed to achieve the desired control, whereas the last two were successful. This insight indicates that the total number of conditional iterations, $J \times M$, is crucial for control effectiveness, given that the first two experiments had a total of 20, while the last two had 100. To achieve the desired results, a higher total count of conditional correction seems to be necessary.

## 4.2 Alternatve Maximization

We shift focus from the probabilistic details of $p(\mathbf{z}_0)$ and $p(\mathbf{y}|\mathbf{z}_0)$ in the following sections, treating them as functions of $\mathbf{z}_0$ under a given condition $\mathbf{y}$. We refer to $p(\mathbf{y}|\mathbf{z}_0)$ as $p_{\mathbf{y}}(\mathbf{z}_0)$, recognizing that the local maxima of $p_{\mathbf{y}}(\mathbf{z}_0)$ are contained within those of $p(\mathbf{z}_0)$. The conditional generation aims to maximize $\log p(\mathbf{z}_0, \mathbf{y})$ by sequentially optimizing $\log p(\mathbf{z}_0)$ and $\log p_{\mathbf{y}}(\mathbf{z}_0)$. This strategy, as outlined in the proposition 1, efficiently optimizes the likelihood $\log p(\mathbf{z}_0, \mathbf{y})$.

**Proposition 1** (Convergence of Alternative Maximization). *Let $A(\mathbf{z})$ and $B(\mathbf{z})$ be two functions defined on the same domain. Suppose that: $S_B$, the local maxima point set of $B(\mathbf{z})$, is a subset of $S_A$, the local maxima point set of $A(\mathbf{z})$. Then, the alternating maximization of $A(\mathbf{z})$ and $B(\mathbf{z})$ converges to a local maximum of the function $A(\mathbf{z}) + B(\mathbf{z})$.*

The detailed proof are reserved for Appendix A. Here, we provide an intuitive explanation for why the proposition holds true: Since the local maxima of $B(\boldsymbol{z})$ are a subset of those of $A(\boldsymbol{z})$, maximizing $B(\boldsymbol{z})$ will not conflict with the maximization of $A(\boldsymbol{z})$, as both functions share the same maxima. In each step of alternating maximization, either $A(\boldsymbol{z})$ or $B(\boldsymbol{z})$ is maximized, ensuring that the combined function $A(\boldsymbol{z}) + B(\boldsymbol{z})$ is always non-decreasing. This process continually improves or maintains the value of $A(\boldsymbol{z}) + B(\boldsymbol{z})$, progressively guiding the optimization towards the shared local maxima. Therefore, alternating maximization converges to a local maximum of the combined function.

Equations (10)(11), along with Equation (12), serve as maximization solvers for $\log p(\mathbf{z}_0)$ and $\log p_{\mathbf{y}}(\mathbf{z}_0)$. The alternative maximization sampling process is presented in Algorithm 1. Notably, lines 9, 10 and 11 of Algorithm 1 ensure that the gradient ascent for $\log p_{\mathbf{y}}(\mathbf{z}_0)$ is always performed on the natural image manifold $M_0$ defined by the diffusion model.

## 5 AccCtr: Accelerating Training-free Conditional Diffusion

Algorithm 1 outlines our framework for training-free CDMs. We will examines the impact of the total iterations $J$, the iterations $N$ for maximizing $p(\mathbf{z}_0)$ (green), and $M$ for maximizing $p_{\boldsymbol{y}}(\mathbf{z}_0)$ (yellow) in the algorithm. Understanding their effects is crucial, as the iteration number significantly influence algorithm performance in training-free CDMs.

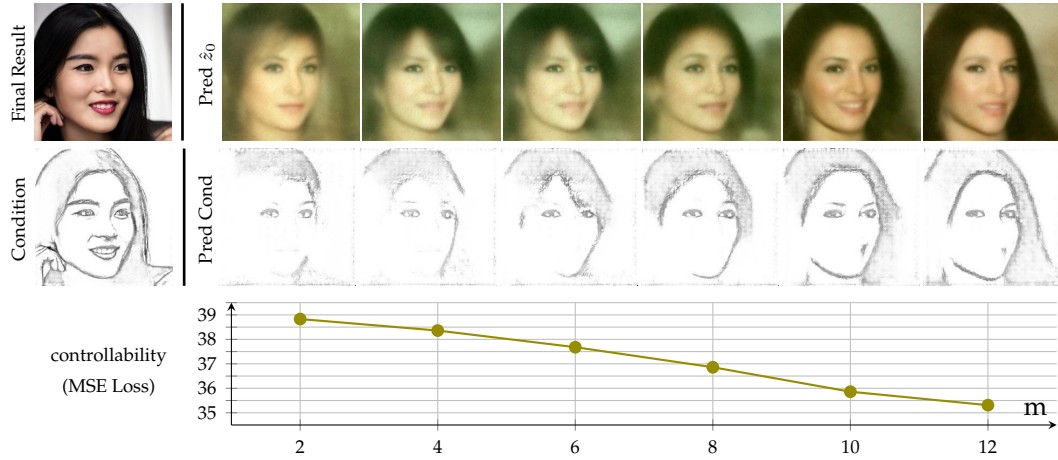

Figure 2: Evolution of Extracted Conditions Across Intermediate Results $\hat{\mathbf{z}}_{0|t}^{(m)}$ of Algorithm 1 at $J = 16$ step with $N = 1$. As the conditional correction count $m$ increases from 2 to 12, the generated results in the first row progressively approximate the final outcome, and the extracted conditions in the second row become more akin to the guidance image. Correspondingly, the MSE plot in the last row exhibits a decreasing trend.

### 5.1 WHY TRANING-FREE CDMs SAMPLING IS SLOW?

Accelerating the sampling speed requires reducing inference steps. The variation in sampling methods often obscures the root causes of this slowness. Proposition 1 helps break down the sampling process into two phases: maximizing $\log p(\mathbf{z}_0)$ via unconditional diffusion and maximizing $\log p_{\mathbf{y}}(\mathbf{z}_0)$ through conditional correction. By integrating existing algorithms into the framework detailed in Appendix B, we can identify the phase that slows down the sampling process.

As depicted in Figure 1, we have conducted four experiments. The first section outlines the condition $\mathbf{y}$, with each row corresponding to a different CDMs and showing performance under $\mathbf{y}$. Four experiments were tested to ensure consistent behavior across methods.

**Experiment 1:** With $J = 20$, $N = 1$, and $M = 1$, 20 iterations were allocated to maximize both $\log p(\mathbf{z}_{0|t})$ and $\log p_{\mathbf{y}}(\mathbf{z}_{0|t})$. Results are in the first section of Figure 1.

**Experiment 2:** Here, $J = 20$, $N = 5$, $M = 1$, with 100 maximization iterations for $\log p(\mathbf{z}_{0|t})$ and 20 steps for $\log p_{\mathbf{y}}(\mathbf{z}_{0|t})$. Results are in the second section of Figure 1.

**Experiment 3:** With $J = 20$, $N = 1$, $M = 5$, 20 iterations were allocated to maximize $\log p(\mathbf{z}_{0|t})$ and 100 steps to $\log p_{\mathbf{y}}(\mathbf{z}_{0|t})$. Results are in the third section of Figure 1.

**Experiment 4:** We set $J = 100$, $N = 1$, $M = 1$, resulting in 100 iterations for both $\log p(\mathbf{z}_{0|t})$ and $\log p_{\mathbf{y}}(\mathbf{z}_{0|t})$. Results are in the second section of Figure 1.

Figure 1 shows that the first two experiments lacked control, but the last two were successful. A higher conditional correction iterations $J \times M$ is key for control, with early experiments at 20 and later at 100. Reducing $\log p(\mathbf{z}_{0|t})$ iterations is okay, yet cutting $\log p_{\mathbf{y}}(\mathbf{z}_{0|t})$ iterations harms sample quality by lessening conditional control.

To clarify why reducing the maximization steps for $\log p_{\mathbf{y}}(\mathbf{z}_{0|t})$ is inadvisable, we conducted **Experiment 5** monitors the progression of the extracted condition from the intermediate outputs $\hat{\mathbf{z}}_{0|t}^{(m)}$, as generated by Algorithm 1 for varying $m$. Figure 2 demonstrates that with the increment of $m$, the extracted condition progressively aligns with the target. Additionally, we employed MSE loss (Sara et al., 2019) to assess the divergence between the intermediate edge condition and the target edge image. The bottom row of Figure 2 illustrates that the MSE diminishes with the growth of $m$, signifying improved conformity to the guidance.

These findings indicate that decreasing the conditional correction count $M$ may result in a loss of control over the final output, as the intermediate conditions could stray from the target. The crux of the issue is the linear manifold assumption, where gradient descent uses the tangent space to approximate the local image manifold. If the starting point is remote from the target, additional linear manifolds are necessary to approximate the intervening region. Therefore, increasing the number of iterations for conditional correction is crucial for navigating the manifold's curvature and obtaining a sample that closely matches the target condition.

## 5.2 Our approach

For pre-trained condition extraction networks $C_{\psi}(\cdot)$, our five experiments suggest that the gradient descent algorithm requires more iterations. This is due to the fact that the gradient $\nabla_{\mathbf{z}_{0|t}}\mathcal{E}(\mathbf{y}, \mathbf{z}_{0|t}, C_{\psi})$ may not provide accurate estimates for large steps. To reduce the number of maximization steps needed for $\log p_{\mathbf{y}}(\mathbf{z}_{0|t})$, we propose to refine the condition extraction network $C_{\psi}(\cdot)$ to improve its accuracy, ensuring that the gradient of $\mathcal{E}(\mathbf{y}, \mathbf{z}_{0|t}, C_{\psi})$ offers a more precise direction for larger steps. Consequently, it is logical to retrain the network $C_{\psi}(\cdot)$ with two distinct objectives:

**The 1st term:** $\mathcal{L}_1(\mathbf{y}, \mathbf{z}_{0|t}, C_{\psi})$ is to effectively extract necessary conditions from $\mathbf{z}_t$. Here, $\mathbf{z}_{0|t}$ represents the projection of $\mathbf{z}_t$ onto the manifold $M_0$.

**The 2st term:** $\mathcal{L}_2(\mathbf{y}, \mathbf{z}_0, \mathbf{z}_{0|t}, C_{\psi})$ is to adjust the gradient of the first term so that it provides accurate directional guidance for larger steps.

The first loss term can be constructed using two distinct strategies. The initial approach, employed by previous training-free CDMs, is defined as $\mathcal{L}_1(\mathbf{y}, \mathbf{z}_{0|t}, C_{\psi}) = \|\mathbf{y} - C_{\psi}(D(\mathbf{z}_{0|t}))\|_2^2$. Here, $D$ is the decoder that converts $\mathbf{z}_{0|t}$ into an image, and $C_{\psi}$ is the pre-defined network for tasks like segmentation, depth mapping, or HED. Typically, these pre-defined networks are substantial, leading to high fine-tuning costs. Moreover, MSE loss may not be suitable for all types of losses; for instance, cross-entropy loss is more fitting for segmentation guidance. In this paper, we propose shifting the similarity comparison from the pixel domain to the latent domain, as shown in Equation 13, where $E$ is the encoder that translates an image into its latent representation. This approach offers two benefits: 1) it allows us to use MSE loss for various guidance types, and 2) it enables us to leverage the same backbone for different condition extraction tasks. Here, we utilize the U-Net architecture from stable diffusion (Rombach et al., 2022a) to handle all guidance tasks.

$$\mathcal{L}_1(\mathbf{y}, \mathbf{z}_{0|t}, C_{\psi}) \coloneqq \left\| E(\mathbf{y}) - C_{\psi}(\mathbf{z}_{0|t}) \right\|_2^2 \tag{13}$$

The second loss term is crafted to fine-tune the gradient for larger steps, aiming to achieve the final outcome in a single iteration. Incorporating $\mathcal{E}(\mathbf{y}, \mathbf{z}_{0|t}, C_{\psi}) = \|E(\mathbf{y}) - C_{\psi}(\mathbf{z}_{0|t})\|_2^2$, we employ the conditional score function $\nabla_{\mathbf{z}_t} \log p(\mathbf{z}_t, \mathbf{y}) = \nabla_{\mathbf{z}_t} \log p(\mathbf{z}_t) + \nabla_{\mathbf{z}_t} \log p_{\mathbf{y}}(\mathbf{z}_t)$ with $\nabla_{\mathbf{z}_t} \log p_{\mathbf{y}}(\mathbf{z}_t) = \sqrt{\bar{\alpha}_t}\nabla_{\mathbf{z}_{0|t}} \log p_{\mathbf{y}}(\mathbf{z}_{0|t})$ to replace the score function in Equation 5. This adjustment ensures that the gradient is more accurately aligned for larger steps. Consequently, we obtain:

$$\mathcal{L}_2(\mathbf{y}, \mathbf{z}_0, \mathbf{z}_{0|t}, C_{\psi}) = \left\| \mathbf{z}_0 - \frac{\mathbf{z}_t + (1 - \bar{\alpha}_t)s(\mathbf{z}_t)}{\sqrt{\alpha_t}} - \lambda(1 - \bar{\alpha}_t)\nabla_{\mathbf{z}_{0|t}}\mathcal{L}_1(\mathbf{y}, \mathbf{z}_{0|t}, C_{\psi}) \right\|_2^2 \tag{14}$$

In this work, we adopt the two loss terms to retrain the condition extraction network $C_{\psi}(\cdot)$, which is subsequently integrated into Algorithm 1. Recognizing that $\mathbf{z}_{0|t}$ is deducible from $\mathbf{z}_t$ through Equation 5 and that $\mathbf{z}_t$ is retrievable from $\mathbf{z}_0$ via Equation 4, we can efficiently train the condition extraction network $C_{\psi}(\cdot)$ with the mere acquisition of the pair $(\mathbf{y}, \mathbf{z}_0)$.

## 6 Experiment

In this section, we conduct thorough experiments and comparisons to showcase the efficacy and strengths of our AccCtr sampling approach, while also providing a detailed account of the experimental configuration.

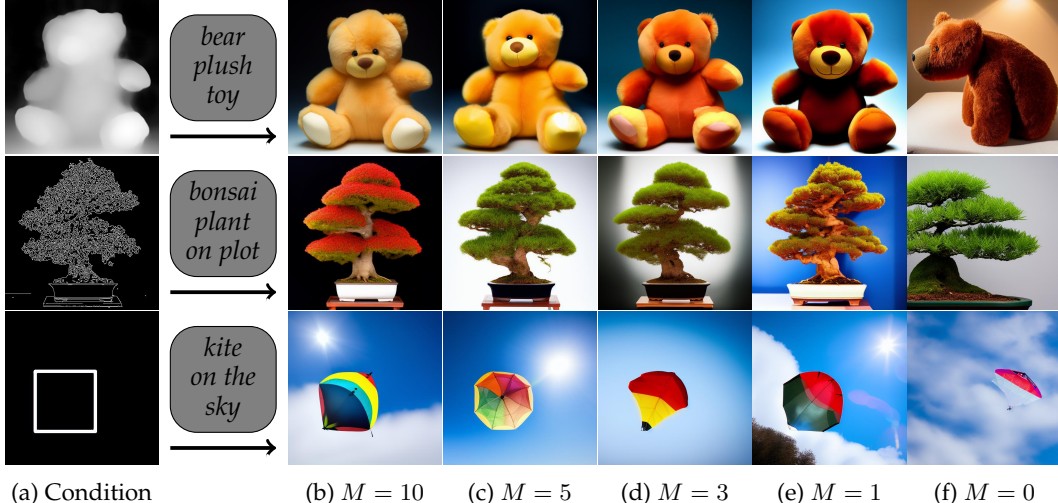

(a) Condition     (b) $M = 10$     (c) $M = 5$     (d) $M = 3$     (e) $M = 1$     (f) $M = 0$

Figure 3: Visual Quality Assessment of Generated Images Across Various Conditional Correction Counts $M$ and Guidances. The first column presents various guidances. The second column lists the prompts. Columns three to seven display the generated images for different values of $M$ with $J = 20, N = 1$.

Table 1: Quantitative Running Cost Comparison. We specify the unconditional diffusion count $N$, conditional correction counts $M$, and sampling time in this table. It is clear that our method provides the fastest outcomes.

|  | UGD | FreeDom | DSG | Ours |
|---|---|---|---|---|
| Unconditional Diffusion Count $N$ (Times) | 500 | 100 | 100 | **20** |
| Conditional Correction Counts $M$ (Times) | 3000 | 90 | 90 | **20** |
| Total Sampling Time (Second) | 2357 | 83 | 53 | **8** |

## 6.1 Implementation Details

We employed the SD-V1.5 model as the foundational backbone for our approach. Our conditional control network closely aligns with the SD-V1.5 model in terms of parameter configuration. To facilitate the training process, we selected the Adam optimizer and set its learning rate to $1e - 5$. With a batch size of 1, the model was subjected to $200,000$ training steps, lasting roughly $60$ hours. In our experiments, we relied on the extensive COCO2017 dataset (Lin et al., 2014), which encompasses approximately $110,000$ images, providing a robust dataset for object detection and segmentation tasks.

## 6.2 Illustrating Sampling Acceleration

In this section, we explore the acceleration capabilities of AccCtr. Proposition 1 suggests that training-free CDMs can be distilled into the optimization of two key objectives. Our experimental results indicate that while the maximum number of iterations for the unconditional objective can be significantly reduced, the same cannot be said for the conditional diffusion, which requires a higher number of iterations. To address this, AccCtr proposes retraining the condition extraction networks $C_\psi(\cdot)$ to decrease the number $M$ of conditional correction iterations needed for the conditional objective $\log p_\mathbf{y}(\mathbf{z}_{0|t})$.

Figure 3 presents the visual quality of images generated by AccCtr for different values of $M$. It's evident that our method can achieve satisfactory results even at $M = 1$, potentially greatly enhancing the sampling speed for CDMs. When $M = 0$, the sampling process does not incorporate conditional control, resulting in outputs that are unaffected by the guidance. Therefore, setting $M = 1$ represents the quickest scenario for conditional generation. To

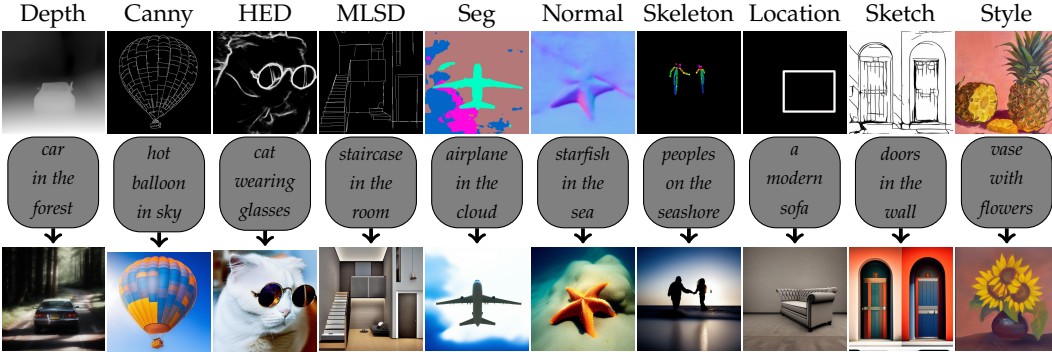

Figure 4: Compatibility Demonstration of MSE Metric for Diverse Guidance Types Using our Condition Extraction Network. We present 10 distinct guidances and their corresponding generated results in this section. Regardless of the variance in guidance, we opt for the same MSE metric to calculate the gradient of $\mathcal{E}(\mathbf{y}, \mathbf{z}_{0|t}, \boldsymbol{C}_{\boldsymbol{\psi}})$.

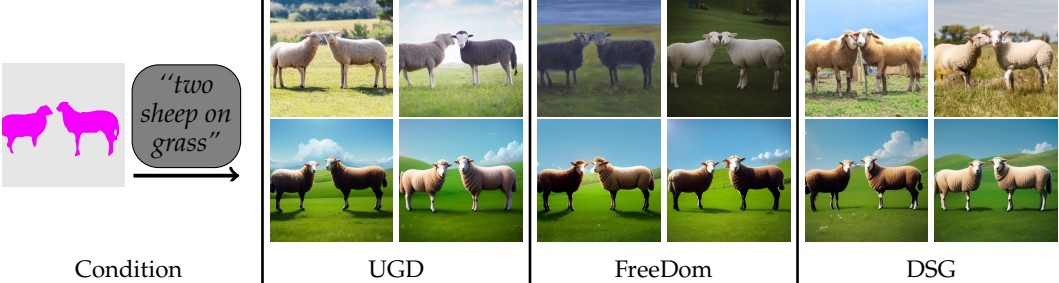

Figure 5: Compatibility Demonstration of our Condition Extraction Network in Conditional Generation Across Different Methods. We have replaced the pre-defined condition extraction networks used by UGD, FreeDoM, and DSG with our own networks. The resulting generated images are displayed in the second row, while originals are in the first.

offer an overview of the acceleration capabilities of our method, we present a quantitative comparison of the running costs in Table 1. We specifically evaluate our method against FreeDoM (Yu et al., 2023), DSG (Yang et al., 2024b), and UGD (Bansal et al., 2024) with respect to the iteration number $N$ for unconditional diffusion, the iteration number $M$ for conditional correction, and the total sampling time. It can be observed that our method incurs the lowest running costs in Table 1.

### 6.3 INVESTIGATING THE COMPATIBILITY OF CONDITION EXTRACTION NETWORKS

In Section 5.2, we highlighted that our condition extraction network can assess the similarity between the guidance and intermediate results using the MSE metric. This approach is notably different from previous methods that employed different metrics for different guidances. Figure 4 displays the visualization results with different guidances, where the similarity is consistently measured using MSE. The results substantiate the compatibility of condition extraction networks for diverse guidances.

Replacing existing pre-defined condition extraction networks with ours is viable, as shown in Figure 5 for FreeDoM (Yu et al., 2023), DSG (Yang et al., 2024b), and UGD (Bansal et al., 2024). The first row shows original results, and the second row shows results with our networks. The sampling quality is comparable, proving our network's compatibility. More importantly, it is potential to accelerate sampling as our network could reduce the conditional correction count $M$ to 1.

Table 2: Quantitative Comparison for Controllable Generation. We selected the depth, canny, and segmentation conditions, which are universally provided by various methods. The best results are highlighted in bold.

| | Depth | | | Canny | | | Segmentation | | |
|---|---|---|---|---|---|---|---|---|---|
| | FID↓ | CLIP↑ | MSE↓ | FID↓ | CLIP↑ | SSIM↑ | FID↓ | CLIP↑ | mIoU↑ |
| ControlNet | 19.3954 | 0.2793 | 90.1302 | **17.3429** | 0.2801 | 0.4138 | 22.1217 | 0.2795 | 0.4217 |
| T2I-Adapter | 23.9216 | 0.2913 | 94.9317 | 17.6812 | 0.3011 | 0.3954 | **22.0173** | 0.2995 | 0.2564 |
| ControlNet++ | **18.0139** | 0.2985 | 87.2173 | 20.1487 | 0.3024 | 0.5138 | 24.9371 | 0.2931 | **0.5438** |
| UGD | 23.0034 | 0.2921 | 86.6792 | 21.8452 | 0.3013 | 0.5037 | 23.5437 | 0.2992 | 0.4127 |
| FreeDom | 22.7825 | 0.2879 | 87.1242 | 21.9547 | 0.2987 | 0.4937 | 23.3619 | 0.2965 | 0.3931 |
| DSG | 23.2147 | 0.2856 | 87.5637 | 21.6153 | 0.2961 | 0.5011 | 23.0198 | 0.2938 | 0.3985 |
| Our | 22.4376 | **0.2932** | **86.0179** | 21.3846 | **0.3041** | **0.5217** | 22.9631 | **0.3011** | 0.4018 |

## 6.4 Ablation Study For Training Loss

Our training loss for the condition extraction networks $C_\psi(\cdot)$ is composed of two key terms. In this section, we perform an Ablation Study on these terms to evaluate their individual importance, with the final results presented in Figure 6. It is evident that without $L_1$, controllable generation is possible but requires a greater number $M$ of conditional corrections. In the absence of $L_2$, controllability is compromised, even with a large number of conditional corrections. In contrast, utilizing condition extraction networks trained with both terms results in more satisfactory outcomes.

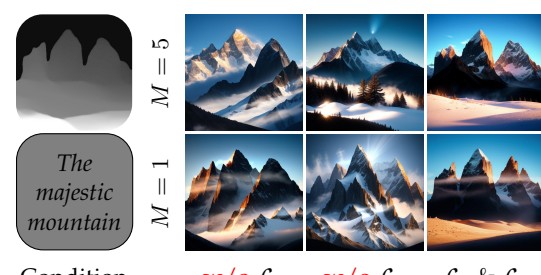

Condition    w/o $\mathcal{L}_1$    w/o $\mathcal{L}_2$    $\mathcal{L}_1$ & $\mathcal{L}_2$

Figure 6: Ablation Study For Training Loss. Each row shows generated results for different $M$. Each column displays the generated results from condition extraction networks trained with various loss configurations.

## 6.5 Sampling Quality Comparison

In this section, we conduct quantitative comparison for sampling quality comparison. Total six methods including three training-free CMDs (FreeDoM (Yu et al., 2023), DSG (Yang et al., 2024b), UGD (Bansal et al., 2024) ) and three training-required CMDs (ControlNet (Zhang et al., 2023), T2I-Adapter (Mou et al., 2024), ControlNet++ (Li et al., 2024) ) are compared. The test is conducted on COCO2017 validation set with timesteps set to 20. For text alignment, we evaluated the CLIP Scores (Radford et al., 2021). For conditional consistency, we measured MSE (Sara et al., 2019) for depth maps, SSIM (Wang et al., 2004) for edge maps, and mIoU (Rezatofighi et al., 2019) for segmentation maps. For conditions not originally supported by training-free CDMs, we have integrated our condition extraction network into their existing algorithms. It is evident that AccCtr leads among pioneering training-free approaches in Table 2, and even when compared to training-required methods, our approach remains competitive. For qualitative comparison, please refer to Appendix C.

## 6.6 Conclusion

Slow sampling is a common issue in current training-free CDMs. In this paper, we introduce a novel framework that reformulates training-free CDMs into the maximization of two distinct objectives. By meticulously counting the optimization steps for each objective, we identify the phase that is the bottleneck for sampling speed and propose retraining the condition extraction networks as a strategy to expedite conditional sampling. Our extensive experiments confirm that AccCtr can significantly reduce the computational cost without compromising sample quality. Most importantly, our method exhibits broad compatibility, holding potential to accelerate a variety of other methods. This conclusion underscores the versatility and efficacy of our approach in addressing the common challenge of slow sampling speeds in training-free CDMs.

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

# A  APPENDIX: PROOF OF PROPOSITION 1

To prove that the alternating maximization of $A(\mathbf{z})$ and $B(\mathbf{z})$ converges to a local maximum of the function $A(\mathbf{z}) + B(\mathbf{z})$, we proceed with the following steps and assumptions.

**Assumptions:**

Let $\mathbf{z} \in \mathbb{R}^n$ denote the variable defined over the domain of the functions $A(\mathbf{z})$ and $B(\mathbf{z})$. We assume:

1. The set of local maxima of $B(\mathbf{z})$, denoted $S_B$, is a subset of the set of local maxima of $A(\mathbf{z})$, denoted $S_A$. That is: $S_B \subseteq S_A$.

2. Both functions $A(\mathbf{z})$ and $B(\mathbf{z})$ are continuously differentiable, and their local maxima are isolated points.

3. The functions $A(\mathbf{z})$ and $B(\mathbf{z})$ have local maxima.

**Alternating Maximization Algorithm:**

The alternating maximization algorithm proceeds as follows:

- Begin with an initial point $\mathbf{z}_0$.
- In each odd iteration (step $k$), maximize $A(\mathbf{z})$, holding $B(\mathbf{z})$ fixed.

$$\mathbf{z}_{k+1} = \arg\max_{\mathbf{z}} A(\mathbf{z}),$$

- In each even iteration (step $k + 1$), maximize $B(\mathbf{z})$ holding $A(\mathbf{z})$ fixed.

$$\mathbf{z}_{k+2} = \arg\max_{\mathbf{z}} B(\mathbf{z}),$$

**Proof:**

We aim to show that this alternating process converges to a local maximum of the combined function $A(\mathbf{z}) + B(\mathbf{z})$.

*Step 1: Local Maxima Relationship*

Suppose at some iteration $\mathbf{z}_k$, we have maximized $A(\mathbf{z})$ so that:

$$\mathbf{z}_k \in S_A.$$

Since $S_B \subseteq S_A$, it follows that if $\mathbf{z}_k$ is also a local maximum of $B(\mathbf{z})$, then:

$$\mathbf{z}_k \in S_B.$$

Thus, at this point, $\mathbf{z}_k$ is a local maximum of both $A(\mathbf{z})$ and $B(\mathbf{z})$.

*Step 2: Behavior of Alternating Maximization*

When we perform alternating maximization, we iterate between optimizing $A(\mathbf{z})$ and $B(\mathbf{z})$. Given the assumption that $S_B \subseteq S_A$, every point that is a local maximum of $B(\mathbf{z})$ is also a local maximum of $A(\mathbf{z})$. Therefore, in each step, when we maximize $B(\mathbf{z})$, the algorithm remains within the set of local maxima of $A(\mathbf{z})$.

As a result, as the algorithm iterates, the points $\mathbf{z}_k$ produced by alternating maximization will always belong to the set $S_A$. Furthermore, the sequence of points $\{\mathbf{z}_k\}$ is confined to a finite set of local maxima (due to the assumption that both functions have finitely many maxima), and the process converges to one of these maxima.

*Step 3: Convergence to a Local Maximum of $A(\mathbf{z}) + B(\mathbf{z})$*

Once the alternating maximization has converged to a point $\mathbf{z}^* \in S_A \cap S_B$, we know that:

- $\mathbf{z}^*$ is a local maximum of $A(\mathbf{z})$
- $\mathbf{z}^*$ is a local maximum of $B(\mathbf{z})$

Because $\mathbf{z}^*$ is a local maximum of both functions individually, it follows that it is also a local maximum of their sum:

$$A(\mathbf{z}) + B(\mathbf{z}).$$

Thus, the alternating maximization process converges to a local maximum of the function $A(\mathbf{z}) + B(\mathbf{z})$.

**Conclusion**

We have shown that the alternating maximization of $A(\mathbf{z})$ and $B(\mathbf{z})$, given the assumption $S_B \subseteq S_A$, converges to a local maximum of the function $A(\mathbf{z}) + B(\mathbf{z})$.

$$\boxed{\text{Q.E.D.}}$$

# B Appendix: Alternative Maximization Sampling Counterpart for FreeDoM, DSG and UGD

Proposition 1 illustrates that conditional sampling is effectively an alternating maximization of two objectives. In this section, we present the Alternative Maximization Sampling framework, which is applied to FreeDoM, DSG, and UGD. The purpose of this framework is to investigate the reasons behind the slow sampling process in training-free Conditional Diffusion Models (CDMs). By leveraging the concept of alternating maximization, we seek to enhance our understanding of the efficiency of these models during sampling. Our analysis reveals that the key differences among these methods lie in their respective corrections for $\hat{\mathbf{z}}_{0|t}^{(m+1)}$. The efficacy of each approach is contingent upon how effectively they adjust the intermediate sample $\hat{\mathbf{z}}_{0|t}^{(m+1)}$ to align with the desired conditional attributes. This insight is pivotal for refining the sampling process and enhancing the overall effectiveness of training-free CDMs. By supplying a more precise correction term, we can reduce the number of optimization steps required.

---

**Algorithm 2** Alternative Maximization Sampling For FreeDoM

---

**Require:** The iteration number $J$, the unconditional diffusion count $N$ for solving $p(\mathbf{z}_{0|t})$ and the conditional correction count $M$ for solving $p_{\boldsymbol{y}}(\mathbf{z}_{0|t})$. The time reversal step $K$.

**Ensure:** $\hat{\mathbf{z}}_{JN} \sim \mathcal{N}(\mathbf{0}, \boldsymbol{I})$, and $\hat{\mathbf{z}}_{0|JN} \leftarrow \sqrt{\bar{\alpha}_{JN}}^{-1}(\hat{\mathbf{z}}_{JN} + (1 - \bar{\alpha}_{JN})s_{\boldsymbol{\theta}}(\hat{\mathbf{z}}_{JN}))$

1: **for** $j = J, \ldots, 1$ **do**
2:     **for** $n = 0, \ldots, N - 1$ **do**
3:         $t \leftarrow jN - n$
4:         $\hat{\mathbf{z}}_{t-1} \leftarrow \frac{\sqrt{\bar{\alpha}_{t-1}}\beta_t}{1-\bar{\alpha}_t}\hat{\mathbf{z}}_{0|t} + \frac{\sqrt{\alpha_t}(1-\bar{\alpha}_{t-1})}{1-\bar{\alpha}_t}\hat{\mathbf{z}}_t + \sqrt{\bar{\beta}_t}\boldsymbol{\epsilon}$
5:         $\hat{\mathbf{z}}_{0|t-1} \leftarrow \frac{1}{\sqrt{\bar{\alpha}_{t-1}}}\hat{\mathbf{z}}_{t-1} + \frac{(1-\bar{\alpha}_{t-1})}{\sqrt{\bar{\alpha}_{t-1}}}s_{\boldsymbol{\theta}}(\hat{\mathbf{z}}_{t-1})$
6:     **end for**
7:     $t \leftarrow (j-1)N$
8:     **for** $m = 0, \ldots, M - 1$ **do**
9:         $\hat{\mathbf{z}}_{K|t}^{(m)} \leftarrow \sqrt{\bar{\alpha}_K}\hat{\mathbf{z}}_{0|t}^{(m)} + \sqrt{(1-\bar{\alpha}_K)}\boldsymbol{\epsilon}$
10:        $\hat{\mathbf{z}}_{0|t}^{(m)} \leftarrow \frac{1}{\sqrt{\bar{\alpha}_t}}\hat{\mathbf{z}}_{K|t}^{(m)} + \frac{(1-\bar{\alpha}_K)}{\sqrt{\bar{\alpha}_K}}s_{\boldsymbol{\theta}}(\hat{\mathbf{z}}_{K|t}^{(m)})$
11:       $\hat{\mathbf{z}}_{0|t}^{(m+1)} \leftarrow \hat{\mathbf{z}}_{0|t}^{(m)} - \lambda\nabla_{\hat{\mathbf{z}}_{0|t}^{(m)}}\mathcal{E}(\mathbf{y}, \hat{\mathbf{z}}_{0|t}^{(m)}, C_{\boldsymbol{\psi}})$
12:     **end for**
13:     $\hat{\mathbf{z}}_{0|t} \leftarrow \hat{\mathbf{z}}_{0|t}^{(M)}$
14: **end for**

---

---

**Algorithm 3** Alternative Maximization Sampling For DSG

---

**Require:** The iteration number $J$, the unconditional diffusion count $N$ for solving $p(\mathbf{z}_{0|t})$ and the conditional correction count $M$ for solving $p_{\boldsymbol{y}}(\mathbf{z}_{0|t})$. The time reversal step $K$.

**Ensure:** $\hat{\mathbf{z}}_{JN} \sim \mathcal{N}(\mathbf{0}, \boldsymbol{I})$, and $\hat{\mathbf{z}}_{0|JN} \leftarrow \sqrt{\bar{\alpha}_{JN}}^{-1}(\hat{\mathbf{z}}_{JN} + (1 - \bar{\alpha}_{JN})\boldsymbol{s_\theta}(\hat{\mathbf{z}}_{JN}))$

1: **for** $j = J, \ldots, 1$ **do**
2:      $t \leftarrow jN - n$
3:      $\hat{\mathbf{z}}_{t-1} \leftarrow \frac{\sqrt{\bar{\alpha}_{t-1}}\beta_t}{1-\bar{\alpha}_t}\hat{\mathbf{z}}_{0|t} + \frac{\sqrt{\alpha_t}(1-\bar{\alpha}_{t-1})}{1-\bar{\alpha}_t}\hat{\mathbf{z}}_t + \sqrt{\bar{\beta}_t}\boldsymbol{\epsilon}$
4:      $\hat{\mathbf{z}}_{0|t-1} \leftarrow \frac{1}{\sqrt{\bar{\alpha}_{t-1}}}\hat{\mathbf{z}}_{t-1} + \frac{(1-\bar{\alpha}_{t-1})}{\sqrt{\bar{\alpha}_{t-1}}}\boldsymbol{s_\theta}(\hat{\mathbf{z}}_{t-1})$
5: **end for**
6: $t \leftarrow (j-1)N$
7: **for** $m = 0, \ldots, M-1$ **do**
8:      $\hat{\mathbf{z}}_{K|t}^{(m)} \leftarrow \sqrt{\bar{\alpha}_K}\hat{\mathbf{z}}_{0|t}^{(m)} + \sqrt{(1-\bar{\alpha}_K)}\boldsymbol{\epsilon}$
9:      $\hat{\mathbf{z}}_{0|t}^{(m)} \leftarrow \frac{1}{\sqrt{\bar{\alpha}_t}}\hat{\mathbf{z}}_{K|t}^{(m)} + \frac{(1-\bar{\alpha}_K)}{\sqrt{\bar{\alpha}_K}}\boldsymbol{s_\theta}(\hat{\mathbf{z}}_{K|t}^{(m)})$
10:      $\boldsymbol{d}^* \leftarrow -\sqrt{n}\sqrt{\bar{\beta}_t}\frac{\nabla_{\hat{\mathbf{z}}_{0|t}^{(m)}}\mathcal{E}(\mathbf{y}, \hat{\mathbf{z}}_{0|t}^{(m)}, \boldsymbol{C_\psi})}{\|\nabla_{\hat{\mathbf{z}}_{0|t}^{(m)}}\mathcal{E}(\mathbf{y}, \hat{\mathbf{z}}_{0|t}^{(m)}, \boldsymbol{C_\psi})\|^2}$
11:      $\boldsymbol{d}^{\text{sample}} = \sqrt{\bar{\beta}_t}\boldsymbol{\epsilon}$
12:      $\boldsymbol{d}_m = \boldsymbol{d}^{\text{sample}} + g_r(\boldsymbol{d}^* - \boldsymbol{d}^{\text{sample}})$
13:      $\hat{\mathbf{z}}_{0|t}^{(m+1)} \leftarrow \hat{\mathbf{z}}_{0|t}^{(m)} + r\frac{\boldsymbol{d}_m}{\|\boldsymbol{d}_m\|}$
14: **end for**
15: $\hat{\mathbf{z}}_{0|t} \leftarrow \hat{\mathbf{z}}_{0|t}^{(M)}$

---

**Algorithm 4** Alternative Maximization Sampling For UGD

---

**Require:** The iteration number $J$, the unconditional diffusion count $N$ for solving $p(\mathbf{z}_{0|t})$ and the conditional correction count $M$ for solving $p_{\boldsymbol{y}}(\mathbf{z}_{0|t})$. The time reversal step $K$.

**Ensure:** $\hat{\mathbf{z}}_{JN} \sim \mathcal{N}(\mathbf{0}, \boldsymbol{I})$, and $\hat{\mathbf{z}}_{0|JN} \leftarrow \sqrt{\bar{\alpha}_{JN}}^{-1}(\hat{\mathbf{z}}_{JN} + (1 - \bar{\alpha}_{JN})\boldsymbol{s_\theta}(\hat{\mathbf{z}}_{JN}))$

1: **for** $j = J, \ldots, 1$ **do**
2:      **for** $n = 0, \ldots, N-1$ **do**
3:          $t \leftarrow jN - n$
4:          $\hat{\mathbf{z}}_{t-1} \leftarrow \frac{\sqrt{\bar{\alpha}_{t-1}}\beta_t}{1-\bar{\alpha}_t}\hat{\mathbf{z}}_{0|t} + \frac{\sqrt{\alpha_t}(1-\bar{\alpha}_{t-1})}{1-\bar{\alpha}_t}\hat{\mathbf{z}}_t + \sqrt{\bar{\beta}_t}\boldsymbol{\epsilon}$
5:          $\hat{\mathbf{z}}_{0|t-1} \leftarrow \frac{1}{\sqrt{\bar{\alpha}_{t-1}}}\hat{\mathbf{z}}_{t-1} + \frac{(1-\bar{\alpha}_{t-1})}{\sqrt{\bar{\alpha}_{t-1}}}\boldsymbol{s_\theta}(\hat{\mathbf{z}}_{t-1})$
6:      **end for**
7:      $t \leftarrow (j-1)N$
8:      **for** $m = 0, \ldots, M-1$ **do**
9:          $\hat{\mathbf{z}}_{K|t}^{(m)} \leftarrow \sqrt{\bar{\alpha}_K}\hat{\mathbf{z}}_{0|t}^{(m)} + \sqrt{(1-\bar{\alpha}_K)}\boldsymbol{\epsilon}$
10:         $\hat{\mathbf{z}}_{0|t}^{(m)} \leftarrow \frac{1}{\sqrt{\bar{\alpha}_t}}\hat{\mathbf{z}}_{K|t}^{(m)} + \frac{(1-\bar{\alpha}_K)}{\sqrt{\bar{\alpha}_K}}\boldsymbol{s_\theta}(\hat{\mathbf{z}}_{K|t}^{(m)})$
11:         $\Delta\hat{\mathbf{z}}_{0|t}^{(m)} = \underset{\Delta}{\arg\min}\, \mathcal{E}(\mathbf{y}, \hat{\mathbf{z}}_{0|t}^{(m)} + \Delta, \boldsymbol{C_\psi})$
12:         $\hat{\mathbf{z}}_{0|t}^{(m+1)} \leftarrow \hat{\mathbf{z}}_{0|t}^{(m)} - \lambda\left(\nabla_{\hat{\mathbf{z}}_{0|t}^{(m)}}\mathcal{E}(\mathbf{y}, \hat{\mathbf{z}}_{0|t}^{(m)}, \boldsymbol{C_\psi}) - \sqrt{\frac{\alpha_t}{1-\alpha_t}}\Delta\hat{\mathbf{z}}_{0|t}^{(m)}\right)$
13:      **end for**
14:      $\hat{\mathbf{z}}_{0|t} \leftarrow \hat{\mathbf{z}}_{0|t}^{(M)}$
15: **end for**

---

## C  APPENDIX: QUALITATIVE COMPARISON

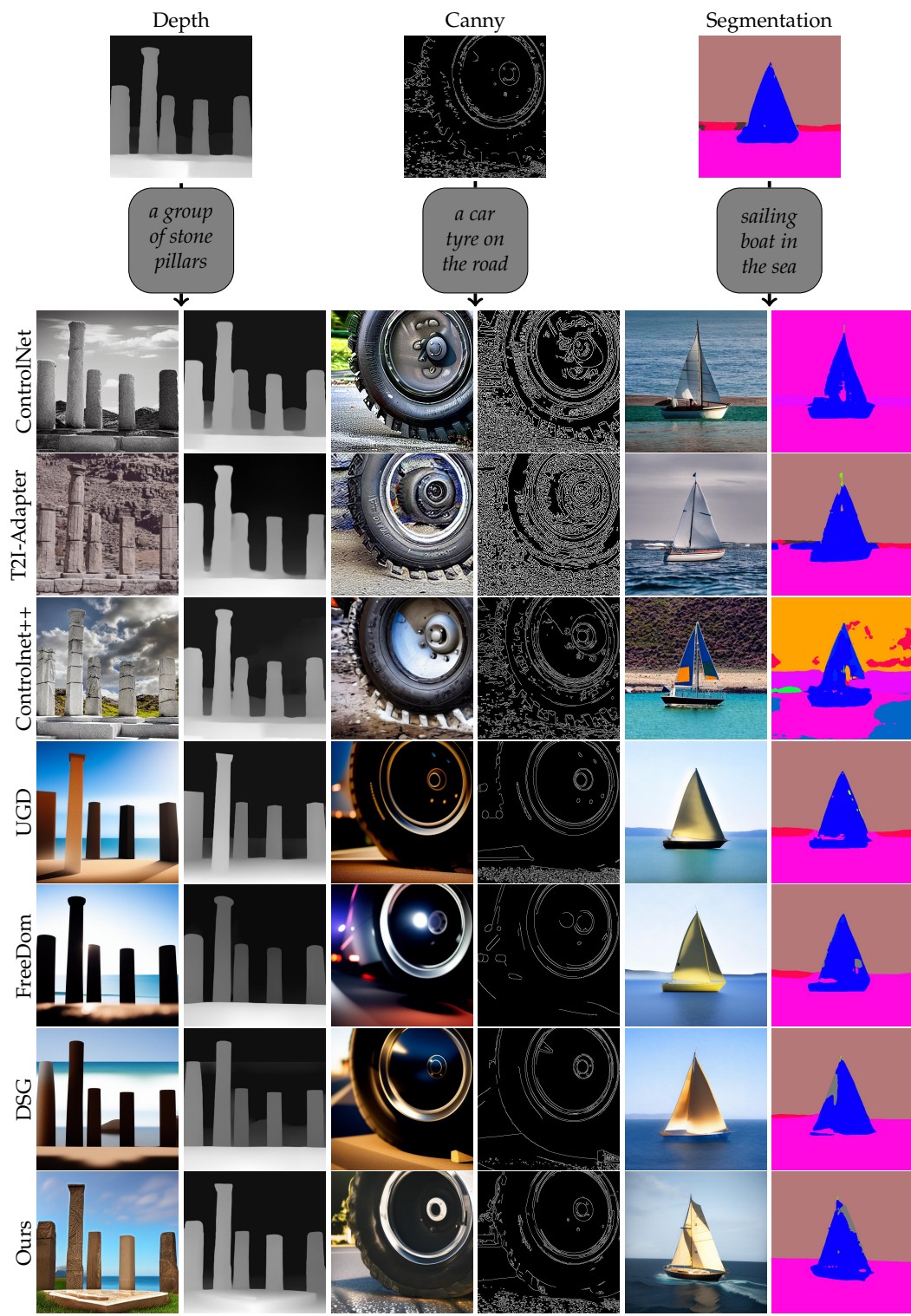

Figure 7: Visual Quality Comparison. In each pair of columns, the first column showcases the generated results, while the second column displays the extracted conditions from these results. It is evident that our method adheres precisely to the guidance compared to other methods.

