# OpenReview forum: "AccCtr: Accelerating Training-Free  Control For Text-to-Image Diffusion Models"
_ICLR.cc/2025/Conference — ICLR 2025 Conference Withdrawn Submission_

### Official Review · Reviewer_ZezA · 2024-10-29

**Soundness:** 2
**Presentation:** 1
**Contribution:** 2
**Rating:** 3
**Confidence:** 4

**Summary:**

This paper aims at training-free conditional generation tasks for the pre-trained diffusion models. Specifically, this paper first introduces alternative maximization to investigate why training-free conditional generation always needs more sampling steps. Then, this paper argued that the reason is that the feature extractor cannot offer precise direction. To solve this question, this paper further proposes how to fine-tune the feature extractor, which could both reduce the sampling steps and enhance the generation quality.

**Strengths:**

1. Introducing the alternative maximization to explore the reason for the slow generation of training-free conditional generation is interesting. Training-free conditional generation is similar to gradient optimization; thus, it is reasonable to introduce the maximization view.

2. The experimental results show that AccCtr achieves the SOTA results based on specific tasks.

**Weaknesses:**

1. The writing for this paper should be further improved. 1) Redundant and confused math equations. For example, the main content in Sec 3.1 is the background of the DDPM, which will never be used again in the latter storyline. Meanwhile, $\tilde{\beta}$ should be $\bar{\beta}$ in 139 lines.  This also leads to difficulty in understanding Algorithm 1. For example, the time travel step $K$ is shown in 9-13 lines in Algorithm 1. We assume $K=2$. We can find that the equation in 9 lines is wrong since the noise scheduler parameter $\bar{a}\_{k}$ will become $\bar{a}\_{2}$, which mismatches the $\bar{a}\_{JN}$ in the outer loop. In the end, what is $z_{0|t}^{m}$ mean in Fig. 2? 2) There is an isolated storyline. There seems to be no connection between Sec. 4 and Sec. 5. 3) Typos. For example, in Table 1, the best CLIP metric should be ControlNet instead of the bold part.

At least, the authors provide a brief explanation of how the DDPM background relates to their method, clarify the notation used in Algorithm 1, and explain the meaning of $z_{0|t}^{m}$ in the figure caption.

2. Part of the contribution of this paper seems limited and unreasonable. There are two contributions to this paper. 1) Introducing alternative maximization. 2) Proposing a fine-tuned strategy for the feature extractor based on the two loss functions.

Focusing on the contribution 2, one of the loss $\mathcal{L}\_{1}$ is from MPGD [1] directly. Concretely, MPGD raised that we could map $z_{t}$ to low-dimension latent space to help the training-free conditional generation. Therefore, MPGD proposed to train an auxiliary autoencoder. Thus, $\mathcal{L}\_{1}$ is the same as the MPGD by replacing the autoencoder with a feature extractor, which weakens the contribution 2. Meanwhile, the other loss $\mathcal{L}\_{2}$ is unreasonable. If it is correct, to optimize $\Psi$, we have to compute a Hessian matrix when calculating $\nabla_{z_{0|t}}\mathcal{L}\_{1}$ since we have to first calculate the partial for $z_{0|t}$ and then calculate the partial for $\Psi$. In this condition, the computation cost is too high, which makes it unreasonable.

At least, the authors should clarify how AccCtr differs from or improves upon MPGD, particularly regarding $\mathcal{L}_1$.

3. The experimental results are not enough. 1) As illustrated above, fine-tuning strategy is similar to training an autoencoder in MPGD. The MPGD-z (MPGD with autoencoder) should be considered as the baseline. 2) Lacking the ablation study for $\lambda$. $\lambda$ is very important for the training-free conditional generation since it directly decides the generation quality. Meanwhile, in this paper, $\lambda$ is also a parameter in $\mathcal{L}\_{2}$. Thus, we have to discuss its influence on the quality of generation.


[1] Manifold Preserving Guided Diffusion. Yutong He and Naoki Murata and Chieh-Hsin Lai and Yuhta Takida and Toshimitsu Uesaka and Dongjun Kim and Wei-Hsiang Liao and Yuki Mitsufuji and J Zico Kolter and Ruslan Salakhutdinov and Stefano Ermon. ICLR 2024.

**Questions:**

1. What is the connection between Fig. 1 and the text-to-image-based tasks?  Unlike tasks in Fig. 1, text-to-image-based training-free conditional tasks face additional challenges from the prompt. That is, we have to deal with the guidance from the text and the guidance from the additional conditions such as depth maps. Therefore, tasks in Fig. 1 lose guidance from the text, which leads to the explanation of the slow generation in Sec. 5 being inconvincible. The author should clarify this question carefully.

2. Does AccCtr work on the style-transfer task? The motivation for this question is that previous works, such as MPGD and FreeDom, could work for the style-transfer task, where we could give a style image as the condition to guide the stable diffusion to generate images that both obey the style and the prompt we provide.

3. Is there any way to decrease the fine-tuning cost? Currently, the fine-tuning cost reaches 60 hours, and the proposed method cannot drop the feature encoders different from the MPGD since MPGD works well, too, after dropping the autoencoder. In this condition, the computation cost is unacceptable for the training-free method.

To sum up, all my concerns are listed in Weaknesses and Questions. The advantage of this paper is that it introduces alternative maximization. However, this paper should be improved further. In this condition, we rate it as "reject." If the author could clarify these concerns, i am wiling to increase my rate.

---

### Official Review · Reviewer_YGcm · 2024-11-01

**Soundness:** 2
**Presentation:** 2
**Contribution:** 2
**Rating:** 5
**Confidence:** 3

**Summary:**

The paper proposes an alternative maximization problem for conditional diffusion models. The key idea seems to be to run an unconditional diffusion chain first, and then, follow up with a conditional correction with a retrained condition extraction network. The paper claims to have improved both the sample quality and sampling time w.r.t. prior art. The approach validated using multiple input modalities (e.g., depth, canny edges, or segmentation masks) and show promising results when compared to prior models.

**Strengths:**

The topic is of interest for the ICLR community.

The figures are easy to read and understandable.

Alg. 1 adds to method understanding.

**Weaknesses:**

The writing of the paper is not easy to follow, it requires multiple readings of the abstract and intro to partially grasp the paper content. The presentation of the methodology section could be modified and simplified. Results section is missing some details, e.g. discussion of Table 2 is very short.

The validation is limited to only one generative model; quantitative results are limited. Adding more generative models (including flow matching ones) as well as adding more benchmark comparisons would make the paper stronger.

Given that the paper claims improvements in model sapling efficiency it would be nice to discuss connections to efficient sampling literature, e.g. https://arxiv.org/abs/2211.13449 and https://arxiv.org/abs/2310.19075

Given the limited validation and lack of positioning to methods that sample efficiently from the diffusion models, the impact of the introduced solution is unclear.

**Questions:**

Please strengthen the paper presentation by improving positioning to related work.

Adding more generative models and quantitative results would strengthen the paper.

In Table 2 the reported differences across methods are not too high. Would it be possible to add confidence intervals to the numbers?

The use of \cite and \citep in the intro is a bit confusing, e.g., in second paragraph all the citations should be with \citep.

---

### Official Review · Reviewer_MZb9 · 2024-11-01

**Soundness:** 3
**Presentation:** 3
**Contribution:** 3
**Rating:** 5
**Confidence:** 3

**Summary:**

To accelerate the sampling speed in text-to-image controllable generation, this paper retrains the condition extraction network to refine the loss's guidance, denoted as the AccCtr framework.This is because the analysis reveals that manifold deviation is a key factor contributing to slow sampling, necessitating more iterations to match the target conditions and the data manifold. In practice, the resulting AccCtr can be seamlessly implemented into current training-free conditional diffusion models with negligible sampling overhead. Importantly, experiments enhanced by the optimized condition extraction network demonstrate both effectiveness and high efficiency.

**Strengths:**

1. Slow sampling speed in diffusion models is a well-known issue that has attracted considerable attention from researchers. Consequently, this paper investigates an intriguing problem and presents meaningful analyses.

2. The analysis of manifold deviation is reasonable and serves as a primary motivation for future work, as a .

3. The proposed alternative maximization is theoretical guaranteed, which provide some inspirations for computer vision task.

4. The presentation is clear, the figures are visually appealing, and the writing is well done.

**Weaknesses:**

1. The connection between the alternative maximization and condition extraction network optimization seems like no strong connection.

2. In my humble opinion, the performance improvements shown in Table 2 are marginal. However, since using an additional network is a common way to enhance generation, it may not be worthwhile to incur extra training costs for only a slight improvement.

3. It seems like make a trade-off between the image quality and controllable force,as the FID value can not compare with other baselines.

4. There is a lack of ablations on the selection of the unconditional diffusion count $N$ and the conditional correction count $M$.

5. It is suggested that the format of the references be made uniform, as there are discrepancies between the introduction and other sections.

**Questions:**

1. Could you provide more experiments in the ablation study with quantitative evaluations, rather than relying solely on visual analyses?

2. Why not include the sampling overhead in Table 2 for a more comprehensive comparison?

3. How about the detailed model architecture of the condition extraction network?

4. In Table 1, why not include some metrics for evaluation when comparing sampling time? This would provide a more reasonable analysis.

If all of my concerns are addressed, I will consider improving the score.

---

### Official Review · Reviewer_xaqj · 2024-11-03

**Soundness:** 1
**Presentation:** 2
**Contribution:** 2
**Rating:** 1
**Confidence:** 4

**Summary:**

AccCtr trains condition extraction network on latent space of LDM such that the gradient can guide posterior means $\mathbb{E}[z_0|z_t]$ toward clean image that is paired with given condition. By applying trained condition extraction network during the diffusion sampling process, AccCtr achieves conditioned sampling.

**Strengths:**

- Training condition extracting network with the proposed objective function provides promising results, without repeating multiple updates on posterior mean $\mathbb{E}[z_0|z_t]$.

**Weaknesses:**

- First of all, the proposed method actually "trains" the conditional network from scratch using paired set of image and generated condition (equation 14), while the title claims that this is method for "training-free" conditioned generation. This could raise confusion for readers so mush be fixed.
- Significant error in the theorem: in equation (12), the paper claims that $\nabla_{x} \mathcal{E}(y, x, C_\psi) = \sqrt{\bar\alpha_{t-1}} \nabla_{f(x)} \mathcal{E}(y, f(x), C_\psi)$. (For simplicity, $x=\hat z_{t-1}$ and $f(x)=(x-\sqrt{1-\bar\alpha_{t-1}}\epsilon_\theta(x,t-1)/\sqrt{\bar\alpha_{t-1}})$). Definitely, the equality does not hold for general function $\mathcal{E}$. Even though one assumes that $\mathcal{E}$ is a linear function, $\partial f(x)/\partial x$ is not a constant $\sqrt{\bar \alpha_t}$, but should contain $\partial \epsilon_\theta(x) / \partial x$. In other words, the conditional score function $\nabla_{z_t} \log p(z_t|y)$ is not equal to $\sqrt{\bar\alpha_t}\nabla_{z_{0|t}} \log p(z_{0|t}|y)$. Thus, the claim (in lines 193-199) that states "equation (10)-(12) alternately maximize the two objective $\log p(z_0)$ and $\log p(y|z_{0|t})$" is invalid and following analysis in section 4-5 is broken.
- In section 3.1, authors claim that $\hat z_{0|t}$ is the projection of $\hat z_t$ on the "image" manifold $M_0$. However, $\hat z_{0|t}$ is posterior mean $\mathbb{E} [z_0|z_t]$ which is conditioned on the time $t$ (i.e. noise level). It would be true that as $t$ goes to 0, manifold of $\hat z_{0|t}$ become closer to $M_0$, but not for large $t$. This could be easily shown by visualizing $\hat z_{0|t}$ during diffusion sampling.
- Regardless of the theorem, the relevant methods that use optimization on Tweedie estimation for conditioned sampling already has been explored in [1,2], but there is no discussion or comparisons with them. The difference of the proposed method is just optimizing updates on denoised estimation $z_{0|t}$ multiple times.
- Missing comparison with training-free methods [3, 4] and adapter-tuning method [5,6].

- Manuscript should be proofread.
	- The notion $z_t$ denotes pixel level diffusion sample in section 1-4 and even for algorithm 1, but it represents latent code in lines 347-350. This would make readers confusing.
	- In equation (9), $\bar\alpha_t$ should be $\bar\alpha_{t-1}$, $\hat z_t$ should be $\hat z_{t-1}$ as equation (11).
	- In section 3.2, $s(z_t|y)$ is more proper for the "conditional" score that $s(z_t, y)$, not just for the definition, but also for the literature [7, 8, 9]. Still, equalities such as $s(z_t|y)=s(z_t)+\nabla_{z_t} \log p(y|z_t)$ hold.

- The style of manuscript does not follow ICLR2025.sty.

**Reference**

[1] Decomposed Diffusion Sampler for Accelerating Large-Scale Inverse Problems (ICLR'24)

[2] DreamSampler: Unifying Diffusion Sampling and Score Distillation for Image Manipulation (ECCV'24)

[3] FreeControl: Training-Free Spatial Control of Any Text-to-Image Diffusion Model with Any Condition (CVPR'24)

[4] MasaCtrl: Tuning-free Mutual Self-Attention Control for Consistent Image Synthesis and Editing (ICCV'23)

[5] ControlNet : Adding Conditional Control to Text-to-Image Diffusion Models (ICCV'23)

[6] Ctrl-Adapter: An Efficient and Versatile Framework for Adapting Diverse Controls to Any Diffusion Model (Arxiv'24)

[7] Score-Based Generative Modeling through Stochastic Differential Equations (ICLR'21)

[8] Diffusion Models Beat GANs on Image Synthesis (NeurIPS'21)

[9] High-Resolution Image Synthesis with Latent Diffusion Models (CVPR'22) - section 3.3

**Questions:**

- In section 5, the submission claims that the gradient may not provide accurate estimates for "large steps". Could authors clarify the meaning of "large steps"? Does it mean large step size for gradient descent?
- What diffusion time $t$ is used for the Figure 2?
- In section 5.2, what is the meaning of "retrain"? It would be great if authors clarify whether it is fine-tuning or training from scratch. More specifically, is the condition extraction network $C_\psi$ trained by two-stage approach with $\mathcal{L}_1$ and $\mathcal{L}_2$? If not, where the authors get the pre-trained condition extraction network (line 334) that extracts condition from latent code of LDM (line 352)?

---

### Note · Authors · 2024-11-14

I have read and agree with the venue's withdrawal policy on behalf of myself and my co-authors.